# Ribosome Profiling and RNA Sequencing Reveal Translation and Transcription Regulation under Acute Heat Stress in Rainbow Trout (*Oncorhynchus mykiss*, *Walbaum*, *1792*) Liver

**DOI:** 10.3390/ijms25168848

**Published:** 2024-08-14

**Authors:** Guiyan Zhao, Zhe Liu, Jinqiang Quan, Junhao Lu, Lanlan Li, Yucai Pan

**Affiliations:** Department of College of Animal Science and Technology, Gansu Agricultural University, Lanzhou 730070, China; 18893812949@163.com (G.Z.); jinqiang@gsau.edu.cn (J.Q.); junhaolu@st.gsau.edu.cn (J.L.); lill@st.gsau.edu.cn (L.L.); panyc@st.gsau.edu.cn (Y.P.)

**Keywords:** cold-water fish, high-temperature stress, Ribo-seq, translational efficiency, upstream open reading frames

## Abstract

Rainbow trout (*Oncorhynchus mykiss*, *Walbaum*, *1792*) is an important economic cold-water fish that is susceptible to heat stress. To date, the heat stress response in rainbow trout is more widely understood at the transcriptional level, while little research has been conducted at the translational level. To reveal the translational regulation of heat stress in rainbow trout, in this study, we performed a ribosome profiling assay of rainbow trout liver under normal and heat stress conditions. Comparative analysis of the RNA-seq data with the ribosome profiling data showed that the folding changes in gene expression at the transcriptional level are moderately correlated with those at the translational level. In total, 1213 genes were significantly altered at the translational level. However, only 32.8% of the genes were common between both levels, demonstrating that heat stress is coordinated across both transcriptional and translational levels. Moreover, 809 genes exhibited significant differences in translational efficiency (TE), with the TE of these genes being considerably affected by factors such as the GC content, coding sequence length, and upstream open reading frame (uORF) presence. In addition, 3468 potential uORFs in 2676 genes were identified, which can potentially affect the TE of the main open reading frames. In this study, Ribo-seq and RNA-seq were used for the first time to elucidate the coordinated regulation of transcription and translation in rainbow trout under heat stress. These findings are expected to contribute novel data and theoretical insights to the international literature on the thermal stress response in fish.

## 1. Introduction

With frequent heat waves and increasing global temperatures, the aquaculture industry, particularly cold-water fish farming, faces environmental pressures [1,2]. As ectotherms, the physiology of fishes highly depends on water temperature; this makes them exceptionally sensitive to temperature fluctuations. An abnormal increase or decrease in water temperature may adversely affect fish survival and reproduction. Cold-water fishes are at a higher risk of experiencing stress than warm-water fishes because of elevated temperatures. The effects of increased temperature on cold-water fish species are multifaceted, with an effect on water oxygen levels, a key environmental factor for disease outbreaks [3,4,5,6], metabolic rate, food intake rate, growth rate, and other factors [7,8,9]. These factors negatively affect fish ecosystems and economic performance.

Rainbow trout (*Oncorhynchus mykiss*, *Walbaum*, *1792*) is a highly valued cold-water fish that is favored by salmonid aquaculture worldwide [10]. This species is highly regarded for its nutritional benefits and adaptability, making it an important species for both economic and scientific research [11]. Some of these previous studies, including serum biochemistry; gene expression examination at the molecular level; and transcriptional, protein synthesis, and metabolic response analyses, have investigated how heat stress affects rainbow trout. Through these studies, insights into the physiological and biochemical changes in rainbow trout owing to heat stress were obtained [12,13,14,15]. In addition, through multi-tiered analysis, we observed that rainbow trout feeding decreases when water temperatures increase above 20 °C, and the growth rate decelerates. Furthermore, if the water temperature exceeds 24 °C, rainbow trout will stop feeding and could be at risk of death, which makes 24 °C a critical threshold for heat stress in this species [16,17,18,19]. It is observed that 18,623 alternative splicing events have been identified from 9936 genes using RNA transcriptome sequencing technology (RNA-seq) and genomic information in rainbow trout under heat stress [20], and the identified genes are associated with the heat stress response. Another study analyzed the role of heat stress in regulating the acute response of rainbow trout to long non-coding RNAs (lncRNAs) and mRNAs by RNA sequencing [17], revealing 18 key lncRNA–mRNA pairs that play an important role in regulating acute heat stress. Existing studies have not yet delved into the effects of heat stress on translation regulation in rainbow trout. This study aims to fill this gap by investigating how heat stress affects the mechanisms of translation regulation in rainbow trout.

Gene expression, a complex biological cascade, is meticulously regulated by several factors. These regulatory mechanisms cover the entire process, from gene transcription to mRNA degradation and protein synthesis and degradation. Notably, these regulatory processes do not occur simultaneously, with constant adjustments to meet cellular demand, and adapt to changes both within the cell and in the external environment [21,22,23]. In organisms, translation is a key link in protein biosynthesis. Precise translational regulatory mechanisms are crucial for correctly expressing specific proteins at specific time points [24]. Therefore, understanding translational regulatory mechanisms has recently garnered particular interest. Ribosome profiling (Ribo-seq), an innovative deep sequencing technology, has been developed to specifically analyze approximately 30 nt fragments of ribosome-protecting mRNAs, called ribosome footprints (RFs), which can accurately track the ribosome location on the mRNA, providing insights into gene expression regulation during translation [25,26,27]. With the rapid development and widespread application of ribosome sequencing technology, translational regulation plays an essential role in responses to various biotic and abiotic stimuli in different biological species, including humans [28], mice [29], zebrafish [30], fruit flies [31], and yeast [32]. In humans, this method has been used to analyze the landscape of translational alterations under zinc oxide nanoparticle (ZnO NP) treatment, providing potential targets for enhancing the anticancer effects of ZnO NPs [28]. Translational analysis of mouse embryonic stem cells revealed the presence of thousands of strong pause sites and novel translation products [29]. Furthermore, ribosome analysis has been combined with machine learning methods to validate lncRNAs during zebrafish development [30]. These translational regulatory mechanisms promote or inhibit the translation process by controlling the ribosome distribution density and may sometimes trigger code shifting or codon reading interruption. The advantage of ribosomal profiling is that the RF can be sequenced on a large scale, and total RNA can be sequenced in high throughput, facilitating the direct comparison of RNA-seq and Ribo-seq data and the calculation of the translation efficiency (TE) of the entire set of transcripts [33]. A higher TE indicates a higher likelihood of the translation of mRNA into protein, offering valuable information on the possibility of some mRNAs being converted into proteins. This information is essential for understanding how cells respond to stress at the molecular level and how protein production is adjusted to cope with adverse environmental conditions.

This study introduces an advanced approach to analyzing the complex interactions between transcriptional and translational regulation in rainbow trout under heat stress by combining Ribo-seq and RNA-seq. This dual-omics strategy fills a gap in studies on heat stress in rainbow trout, providing a comprehensive view of molecular responses at the transcriptional and translational levels. In addition, it will provide an important scientific basis and technical support for the aquaculture industry to cope with the challenges posed by global climate change.

## 2. Results

### 2.1. Overview of Ribo-seq and RNA-seq

To comprehensively investigate the effect of thermal stress on rainbow trout at the molecular level, liver samples were collected from fish in the 18 °C control group (CG) and the heat stress group (HS) (24 °C for 8 h). RNA-seq and Ribo-seq were performed, with three replicates for control and heat stress groups. Approximately 336 million and 353 million Ribo-seq reads were generated in the CG and HS, respectively (Appendix A). In addition, around 108 million and 140 million RNA-seq reads were obtained in the CG and HS, respectively (Appendix A). First, the raw reads obtained from Ribo-Seq and RNA-seq were preprocessed to remove any low-quality sequences. Moreover, Pearson correlation coefficients (R^2^), which assess the correlation between transcriptional and translational levels, increased from 0.82 to 0.83 under heat stress (Appendix A).

### 2.2. Features of Ribo-seq

To further determine the effects of thermal stress on the features of Ribo-seq information, the fundamental RFs of the CG and HS were compared. The length distribution of most RFs was predominantly between 26 and 28 nt in the CG and HS (Figure 1a). This finding is consistent with that of previous research on yeast [26]. In addition to protein-coding open reading frames (ORFs), several small ORFs have been discovered in the untranslated regions (UTRs) of different eukaryotic species, including mammals [34], yeast [35], and plants [36]. In all, 2.9%, 2.6%, and 0.2% of RFs were located in the introns, the 5′UTR, and the 3′UTR, respectively, with most RFs in the coding sequences (CDSs) and an average distribution ratio of 94.2% in the CG (Figure 1b). Compared with the CG, the distribution ratio of the RFs mapped to the introns, the 5′UTR, and the 3′UTR increased to 3.2%, 2.9%, and 0.3% in the HS, respectively. However, the ratio of the RFs in the CDS region decreased to 93.6% (Figure 1b). The distribution patterns of the RFs observed in this study are consistent with those reported in previous studies [30,37]. This validation further confirms the superior quality of the Ribo-seq library [38,39]. These findings indicate that UTR translation may affect the response of rainbow trout to heat stress. A clear pattern of 3 nt intervals was observed near the start and end codon regions (Figure 1c).

### 2.3. Response to Heat Stress at the Transcriptional and Translational Levels

Next, we investigated heat-stress-induced gene expression changes at the transcriptional and translational levels separately. First, 2880 genes were upregulated and 2165 genes were downregulated at the transcriptional level. Translational analysis revealed that 924 genes were upregulated and 289 genes were downregulated (Figure 2C). Only a few genes (24.5% of the upregulated genes and 8.3% of the downregulated genes) were shared between transcription and translation (Figure 2D), indicating inconsistent changes at the two levels. Visualization using the Interactive Genome Viewer (IGV) browser revealed global tracks of RNA-seq and Ribo-seq for 32 chromosomes, highlighting heat-stress-induced changes at various levels. The genes that were regulated only at the transcriptional or the translational level are in light green, whereas those regulated at both levels are in light red (Figure 2E).

By applying the criteria of |log 2 FC| ≥ 1 and FDR < 0.05, the genes were classified into nine quadrants (Figure 3A). No significant changes in expression were noted in more than 91.88% (38,494) of the genes at both transcriptional and translational levels (quadrant E). Further analyses showed that 2.24% of the responding genes belonged to the corresponding groups (classes C and G) and that their expression was increased or decreased to a similar extent at the transcriptional and translational levels. In addition, the remaining 5.87% (2454) of the genes were inconsistently regulated between the transcriptional and translational levels; these genes were distributed in quadrants A, B, D, F, H, and I.

Next, to clarify the biological functions of the differentially expressed genes (DEGs) between the CG and the HS, Kyoto Encyclopedia of Genes and Genomes (KEGG) analysis was performed to identify the enrichment pathways among the genes. Furthermore, the endoplasmic reticulum protein processing, MAPK, NOD-like receptor, apoptosis, and Toll-like receptor signaling pathways were significantly enriched in quadrant C. Steroid biosynthesis, DNA replication, the cell cycle, and the p53 signaling pathway were enriched in quadrant G (Figure 3B). Notably, at both levels, the genes hsp70 (ncbi_110512845), hsp90a1 (ncbi_110522488), hsp90b1 (ncbi_110500099), TLR5 (ncbi_100135812), C3-like (ncbi_118939581), and IRF6 (ncbi_110497044) were significantly upregulated under heat stress conditions (Figure 3C).

In addition, genes from four discordant regulatory groups were further analyzed for gene ontology (GO) enrichment. Genes in classes D and F were regulated at the translational level, without significant changes in transcription. Among them, 90 downregulated genes in class D were enriched in glycoside biosynthesis, glycoside metabolism, and DNA catabolic processes (Appendix A). Meanwhile, 163 genes upregulated in class F were significantly enriched in protein-folding processes (Appendix A). In contrast, genes in classes B and H were regulated at the transcriptional level but did not change significantly in translation. In class H, 1314 genes with reduced transcription were significantly enriched in catalytic and oxidoreductase activities, small-molecule binding, and ribosome-binding pathways (Appendix A). There were 890 transcriptional upregulated genes (class B) involved in pathways such as acyl phosphatase activity and RNA binding (Appendix A).

#### 2.3.1. Changes in TE in Response to Heat Stress

TE is a crucial indicator of the translation procedure, indicating the effective use of RNA; it is calculated using FPKM_Ribo-seq_/FPKM_RNA-seq_ [26]. The TE analysis of the entire genome revealed extensive and variable translational regulation in many genes, with log2 (TE) values of −6 to 5 (Figure 4B). In the HS, the TE of 809 genes was notably altered, with 368 genes upregulated and 441 genes downregulated compared to the CG (Figure 4C and Appendix A). Only a few genes exhibited significant changes in TE compared with those observed at the transcriptional and translational levels. The genes were categorized into five groups based on the changes in gene TE and transcription levels (Figure 4D). Only 10 genes were regulated at the transcriptional and TE levels, such as the heat shock protein genes (hsp90a1 and hsp70a). A total of 4359 DEGs were exclusively regulated at the transcriptional level, with no regulation at the TE level. The genes included hsp90b1, Hif1A, Casp3, and MAP3K5. These DEGs primarily participated in processes such as folding, sorting, degradation, transport and catabolism, cell growth and death, and genetic-information-processing pathways. In contrast, 343 genes differed significantly only in TE, with no significant differences in the transcriptional level. These genes included hspa4, ATP5PD, hsc70, hspa8, c3, and vtg1, primarily involved in cellular processes, metabolism, signal transduction, transcription, signaling molecules and interaction, and the vitellogenin pathway. Gene ontology (GO) annotation revealed that these DEGs that differed significantly in TE are predominantly clustered in pathways associated with cellular processes, metabolism, biological regulation, responses to stimuli, signaling mechanisms, immune system functions, cellular components, binding interactions, and catalytic activities (Figure 4E).

#### 2.3.2. Effect of uORF Features on TE

Specific gene sequence features affect TE [39]. To explore the specific contribution of sequence features in translational regulation in rainbow trout, we examined three sequence characteristics (sequence length, normalized minimal free energy (NMFE), and GC content) in the 5′UTR, 3′UTR, and CDS of genes. In the HS, for the 5′UTR, genes with higher TE (log2(TE) > 1) tended to have a lower sequence length and a higher GC content compared to −1< log2(TE) ≤ 0 (Figure 5A). For the 3′UTR, genes with higher TE (log2(TE) > 1) tended to have a longer sequence length, a higher GC content, and lower NMFE compared to −1< log2(TE) ≤ 0 (Figure 5B). For the CDS, genes with higher TE (log2(TE) > 1) tended to have a shorter sequence length and higher NMFE compared to −1 < log2(TE) ≤ 0 (Figure 5C). The sequence features of the genes in different TE groups in the HS were consistent with those in the CG (Appendix A).

#### 2.3.3. Identification of uORFs and Its Effect on mORF Translation

uORFs can regulate the translation of downstream main ORFs (mORFs) [40]. However, studies on uORFs in rainbow trout and their association with heat stress are limited. The patterns surrounding the ATG initiation codon of translated and untranslated uORFs were analyzed using the SeqLogo R package. As a result, 3468 uORFs (Figure 6A) were predicted on 2676 genes in rainbow trout (Figure 6B), with an average of 1.3 uORFs per gene.

Next, three metrics associated with mORF translation were examined to compare the characteristics of translated and untranslated uORFs: uORF length, 5′UTR length, and NMFE [41]. The lengths of translated uORFs (*p*-value = 7.74 × 10^−8^, Figure 6C) and the 5′UTR (*p*-value = 5.9 × 10^−4^, Figure 6D) were significantly shorter than those of untranslated uORFs, with the translated group exhibiting a stronger folding potential (*p*-value = 4.84 × 10^−6^, Figure 6E) in the HS. Furthermore, the relative distances from translated uORFs to the start codon of mORFs (*p*-value = 0.06, Figure 6F) and the transcription start site (TSS) (*p*-value = 0.11, Figure 6G) were shorter compared to those from untranslated uORFs. These findings were consistent with those observed in the CG (Appendix A).

The Kozak consensus sequence (GCCA/GCCAUGG) interacts with translation initiation factors to promote the translation initiation of mRNAs with a 5′UTR cap structure. The sites −3(A/G) and +4(G) near the AUG start codon are particularly important for recognizing mRNA and initiating translation, as reported by Ivanov et al. and Kozak [42,43]. The probability of guanine at the −1 position was higher in the translated uORFs than in the untranslated uORFs (Figure 6H), and a higher GC content in the translated uORFs at the −4 to −1 position was observed (Figure 6I). To better comprehend how translated uORFs affect the TE of mORFs in rainbow trout, the changes in TE among three gene categories were detected: genes without translated uORFs, genes with only a single translated uORF, and genes with multiple translated uORFs. No significant difference was noted in the TE between untranslated and single translated uORFs (*p*-value = 0.0768); however, genes with multiple translated uORFs significantly increased the TE of their mORFs compared with single translated uORFs (*p*-value = 2 × 10^−4^, Figure 6J).

## 3. Discussion

Increased temperatures are a significant abiotic factor affecting cold-water fish species’ viability. Investigating the molecular mechanisms by which cold-water fishes respond to thermal stress can improve understanding of their adaptation to environmental changes. With advances in biotechnology, a technology that can build a bridge between mRNAs and proteins, called translational genomics, has garnered attention. This study combined Ribo-seq and RNA-seq analyses to determine the relationship between the transcriptional and translational levels and differences in gene expression and evaluated how uORFs affect the TE of downstream mORFs. Although this study was a fundamental investigation, it may offer foundational principles and valuable scientific knowledge for enhancing the resilience of rainbow trout against heat stress in the future.

Translation plays an important role in multiple cellular functions, such as growth, development, and adaptation to environmental changes, exhibiting significant flexibility and accuracy [44]. Many studies have verified that plants and animals exhibit notable changes in mRNA and protein levels in response to stress [45,46]. Therefore, translational responses and regulation have become a research hotspot. RFs represent the mRNA fragment that is being translated by the ribosome, and the RF length can reflect ribosome distribution on the mRNA, including the TE of ORFs, the density of ribosomes, and the probability of uORFs. This information is essential for understanding the dynamic process of gene expression regulation and protein synthesis [47]. In this study, the ribosome of rainbow trout was noted to be approximately 27 nt in length (Figure 1a); this is consistent with the RF length of Pelteobagrus fulvidraco [48] but comparatively shorter than that of mice (~33 nt) [29] and Arabidopsis thaliana (~30 nt) [49]. This demonstrates differences in the ribosome state in different species.

By successfully acquiring the Ribo-seq and RNA-seq data, the gene expression changes at the genome-wide level and the interactions at the transcriptional and translational levels under heat stress were investigated. Comprehensive analyses showed that the Pearson correlation coefficient between gene expression changes of transcription and translation increased from 0.82 to 0.83 under acute heat stress (Appendix A). Such small changes may reflect long-term trends or cyclical changes. However, these changes may accumulate over time and have significant biological effects. There were 2464 genes that exhibited inconsistent changes in transcription and translation under heat stress (quadrants A, B, D, F, H, and I), which clearly demonstrated independent stress responses at the two levels. Of them, 2203 genes in quadrants B and H were only regulated at the transcriptional level, without any changes at the translational level (Figure 3A). Translation can be a faster and more immediate response to environmental changes because new mRNA need not be produced in this process [50,51]. Thus, although translation is a fine-tuning effect, it apparently plays a relatively independent role in the stress response. However, 253 genes were significantly altered only at the translational level under heat stress (quadrants D and F) (Figure 3A). The primary functions of these genes are associated with protein folding and modification, and endoplasmic reticulum apoptotic pathways. In a previous study in which yeast was treated with sodium chloride, a higher correlation was noted between mRNA abundance at a particular time point and protein abundance at a subsequent time point relative to the same time points. This suggests that protein expression delays transcriptional activity in response to stress [52]. Therefore, we hypothesize that the changes in the mRNA expression in quadrants D and F may serve as a predictor for future translational changes. The 163 genes exclusively upregulated in the F quadrant suggest that rainbow trout can maintain essential processes by adjusting mRNA expression to counteract decreased TE. Another possible explanation is that the increase in mRNA abundance may create a reserve pool that can be used to accelerate the translation process when the stress is relieved [53]. Sudden and severe stress may trigger separate gene reactions at the transcriptional and translational stages, unlike long-term stress, possibly resulting in more simultaneous regulatory processes at both levels. Transcriptional and translational regulations interact, while remaining independent, creating a complex network that boosts the adaptability and flexibility of gene expression in response to temperature variations. The “protein processing in the endoplasmic reticulum” pathway involves protein folding, modification, and transport, and heat shock protein genes are the primary DEGs enriched in this pathway. Cells produce different proteins, called heat shock proteins, in response to different stressors, such as high temperatures [54], starvation [55], and hypoxia [56]. Heat shock proteins function as molecular chaperones, enhancing cellular resistance to heat stress by properly refolding denatured proteins and promoting the repair or degradation of damaged proteins. Consistent with our findings, hsp70 and hsp90 were increased at the transcriptional and translational levels after the exposure of rainbow trout to 24 °C. Therefore, both transcription and translation may coregulate the expression of heat shock proteins, maintaining homeostasis and protecting against heat stress damage. In general, gene expression is precisely regulated by TFs, microRNAs, lncRNAs, etc. In addition, it is affected by its own sequence features. In this study, TE varied with different sequence features in the CG’s and HS’s 3′UTR, 5′UTR, and CDS (Figure 5). Several studies have revealed that the UTR of a gene contains numerous elements involved in translational regulation. uORFs, as brief coding segments, are located in the 5′UTR of mRNA. In eukaryotic organisms, these regions affect gene expression regulation. uORFs significantly affect gene expression by affecting translation and potentially serving as regulatory elements [57,58]. However, inconsistent with previous studies [59], no significant changes were observed to indicate that rainbow trout uORFs affect the TE of mORFs (Figure 6J). Understanding the function and regulation of uORFs is essential for elucidating the intricate mechanisms underlying gene expression in eukaryotic cells. Further research into the specific interactions and molecular pathways involved in mediating gene regulation by uORFs can provide valuable insights into the complexity of regulatory mechanisms at the genetic level. This study comprehensively analyzed the sequence characteristics of translated and untranslated uORFs (Figure 6C-G). In the HS, the length of translated uORFs was significantly shorter than that of untranslated uORFs, suggesting that shorter uORFs are more easily recognized and translated by the ribosome; therefore, they play a more effective role in regulating mORF translation. Similarly, the 5′UTR containing translated uORFs is shorter than that containing untranslated uORFs. A shorter 5′UTR may facilitate faster translation recognition and initiation by the ribosome, streamlining the process and enhancing TE [60]. This analysis highlights distinct differences between translated and untranslated uORFs in terms of their length, folding potential, and relative distance to the mORF start codon and the TSS. Translated uORFs are shorter in length; however, they have a higher folding potential and are located closer to the mORF start codon and the TSS. These findings provide valuable insights into the translational regulatory mechanisms and may offer implications for further research in this field. Furthermore, uORF regulation is affected by various factors, with the conservatism of Kozak sequences and start codons being key factors [61,62]. Kozak sequences are nucleotide sequences located around the start codon that generally take the form of GCCA/GCCAUGG, in which AUG is the start codon. In contrast, the GCC and CCAUGG sequences are located upstream and downstream of it, respectively (Figure 6H). The conservation of this sequence is essential for accurately placing the ribosome and identifying the start codon. Furthermore, we analyzed the occurrences of the nucleotides adjacent to the initiation codon of uORFs in both translated and untranslated regions and found that the GC content of the sequences surrounding the start codon is higher in the translated uORFs (Figure 6I). Translated uORFs can have a higher effect on the translational regulation of subsequent mORFs. Flanking sequences with a high GC content may enhance the TE of uORFs, thereby more effectively competing with the CDS for ribosomes and inhibiting their translation [63]. These findings highlight the diversity and complexity of the mechanisms regulating gene expression and translation in different species and biological contexts. Future studies should adopt a multifaceted approach to fully understand the role of these regulatory mechanisms in gene expression.

This study offers significant contributions to understanding translational regulation in rainbow trout subjected to thermal stress, yet certain limitations accompany it. A primary constraint is the limited sample size used, potentially impacting the broader applicability of the findings. Enlarging the sample size in subsequent research could serve to validate and further elaborate on these outcomes. Additionally, the study’s concentration on a single species, the rainbow trout, may not encapsulate the full spectrum of responses exhibited by other cold-water fish species when faced with thermal stress. It would be advantageous for subsequent studies to incorporate a comparative approach across a variety of fish species to grasp the wider relevance of these findings.

## 4. Materials and Methods

### 4.1. Ethics Statement

All experiments involving rainbow trout were conducted according to the regulations set forth by Chinese law regarding the handling and treating of animals in laboratory settings. The experimental procedures were approved by the Ethics Committee at Gansu Agricultural University (no. GSAU-2019-52).

### 4.2. Heat Stress Challenge and Sample Collection

A total of 300 full-sib healthy (100 ± 5.0 g) rainbow trout were obtained from the Aquatic Science Training Base of Gansu Agricultural University, Gansu Province. Before the formal study, all rainbow trout were randomly divided into two groups in a flow-through water system (3000 L). The rainbow trout in the CG were allowed to acclimatize at a temperature of 18 °C for 2 weeks. However, the rainbow trout in the HS were subjected to acute heat stress for 8 h based on data from a previous study [64]. The water temperature was closely monitored to quickly increase it from 18 °C to 24 °C within 30 min to simulate acute heat stress conditions. Anesthetized with a lethal dose (80.0 mg/L) of MS-222 (Sigma Aldrich Co., St. Louis, MO, USA) to euthanize the experimental fish, liver samples were collected from rainbow trout [65]. Afterward, the liver samples were frozen in liquid nitrogen and stored at −80 °C. Three biological replicates were selected from the many individual liver samples of rainbow trout in the CG and HS treatments for Ribo-seq and RNA-seq.

### 4.3. RNA Extraction, Library Construction, and Sequencing

TRIzol reagent (Invitrogen, NJ, USA) was used to extract total RNA from the liver samples of the CG and HS. The extracted RNA was then used to create RNA-seq libraries. Thereafter, the cDNA libraries were sequenced by Gene Denovo Biotechnology (Guangzhou, China) using the Illumina NovaSeq6000 platform. After removing sequences containing the ambiguous base N, low-quality sequences, and potential duplicates, high-quality sequences (clean reads) were matched with the reference genome (USDA_OmykA_1.1) of rainbow trout using Bowtie2 alignment software (version 2.2.8). DESeq2 software (http://www.r-project.org/) (version 1.20.0) was then used for differential expression analysis to estimate the expression of individual gene transcripts accurately. Genes with absolute fold-change values ≥ 2 and FDR < 0.05 were considered DEGs. Using the GO database (http://geneontology.org) (GO accessed on 2023.1.5) and the KEGG database (http://www.genome.jp/kegg/) (Release 101), the DEGs were analyzed for enrichment of GO terms and enrichment of KEGG pathways, respectively. The results of q-value < 0.05 were significantly enriched.

### 4.4. RF Extraction and Ribo-seq Library Construction

Previously described methods were used to isolate ribosomes [25,66]. Briefly, liver samples from the CG and HS were rapidly ground into powder and dissolved in lysis buffer. After incubation on ice for 10 min, the samples were centrifuged at 17,000× *g* for 10 min at 4 °C, resulting in the retrieval of 400 μL of the supernatant. RFs were produced by combining 10 μL of RNase I (NEB, Ipswich, MA, USA) and 6 μL of DNase I (NEB, Ipswich, MA, USA) with 400 μL of the supernatant and incubating the mixture at 25 °C for 45 min. To terminate the reaction, 10 μL of SUPERase-In RNase inhibitor (Ambion, Austin, TX, USA) was added. Subsequently, RNA fragments longer than 17 nt were isolated using the protocol of the RNA Clean and Concentrator-25 kit (Zymo Research, Beijing, China). After eliminating ribosomal RNA, RFs were further refined using magnetic beads [67]. NEBNext^®^Multiple Small RNA Library Prep was used to generate Ribo-seq libraries, which were subsequently examined on the Illumina HiSeq™X10 platform by Gene Denovo Biotechnology Company (Guangzhou, China).

### 4.5. Ribo-seq Data Analysis

The original Ribo-seq data were subjected to quality control procedures to eliminate poor-quality sequences and ribosomal RNA. The remaining reads with lengths between 10 and 50 bp were then mapped to rRNAs, tRNA, small nuclear RNA (snRNA), and small nucleolar RNA (snoRNA) databases. In addition, unmapped reads with lengths between 25 and 35 bp were retained and mapped to the rainbow trout’s reference genome (USDA_OmykA_1.1) using Bowtie2 with no mismatches. RFs were assigned to different genomic features (5′UTR, CDS, 3′UTR, and others) based on the position of the 5′ end of the alignment. Furthermore, the RF density at different codon positions was calculated to monitor sequencing reliability. The EdgeR toolkit was used to detect DEGs, where genes with fold-change values ≥ 2 and FDR < 0.05 were categorized as DEGs at the translation level. Moreover, the 3 nt periodic distribution model was effectively generated using the RiboWaltz R toolkit [68].

### 4.6. Comparison of the Differences between Translation and Transcription

Genes with FPKM ≥ 1 were used to determine pairwise Pearson correlation coefficients (R^2^) to assess the relationship between gene transcription and translation. Variations in gene expression at both levels were used to categorize the genes into nine quadrants: A (transcription up, translation down), B (no change in translation, transcription up), C (upregulation in both levels), D (no change in transcription, translation down), E (no change in both levels), F (no change in transcription, translation up), G (downregulation in both levels), H (no change in translation, transcription down), and I (transcription down, translation up). TBtools was used to visualize the data in gene expression heat maps [38]. Furthermore, analyses were performed to enrich GO functions and KEGG pathways for genes within the nine categories. Coordinated regulatory genes were defined as those in quadrants C and G, whereas uncoordinated regulatory genes were categorized into the other groups. IGV version 2.8.3 was used to simultaneously illustrate the changes in gene expression at the transcriptional and translational levels.

### 4.7. Correlation of Transcription and Translation

Translational efficiency (TE) represents the proportion of total RNA molecules (usually referred to as mRNA) of a gene in a sample that binds to ribosomes and undergoes translation, and it is an important indicator for describing the RNA translation process. The mean values of the expression of the same gene at the translation and transcription levels in each group of samples were calculated separately, and then the TE of each gene in each group was calculated: TE = FPKM_Ribo-seq_/FPKM_RNA-seq_ [25]. Based on the differences in TE, the genes were classified into four different categories: log2TE ≤ 1, −1 < log2TE ≤ 0, log2TE ≤ 1, and log2TE ≥ 1) [69]. log2(TE) > 0 indicates that TE > 1, the post-transcription product of the gene accumulates, and the RNA is used; log2(TE) < 0 is the opposite effect; log2 (TE) > 1 indicates that TE > 2, which means that the translation is more efficient; and log2 (TE) < −1 is the opposite. The importance of the differences in gene characteristics among the two TE groups was evaluated using a two-tailed Student’s *t*-test at a significance level of 0.05.

### 4.8. Analysis of uORFs

Sequences of the 5′UTR were collected from established protein-coding genes, where uORFs were 60–450 nt long and included an ATG start codon. uORFs with FPKM ≥ 1 were considered translated uORFs. The R package (SeqLogo) (version 1.38.0) enriched the motifs surrounding the ATG start codon in both translated and untranslated uORFs.

## 5. Conclusions

In this study, the molecular dynamics of rainbow trout under heat stress were investigated by performing RNA-seq and Ribo-seq. A small proportion of response genes were shared at the transcriptional and translational levels, which revealed the independence of the rainbow trout transcription and translation recovery response after acute heat stress. Further analysis revealed that the TE of the genes is greatly influenced by their sequence characteristics, including their GC content, CDS length, and NMFE. Moreover, uORFs provide a new direction for research on regulating the heat resistance of rainbow trout. These findings will serve as a strong foundation for future studies on temperature regulation in rainbow trout.

## Figures and Tables

**Figure 1 ijms-25-08848-f001:**
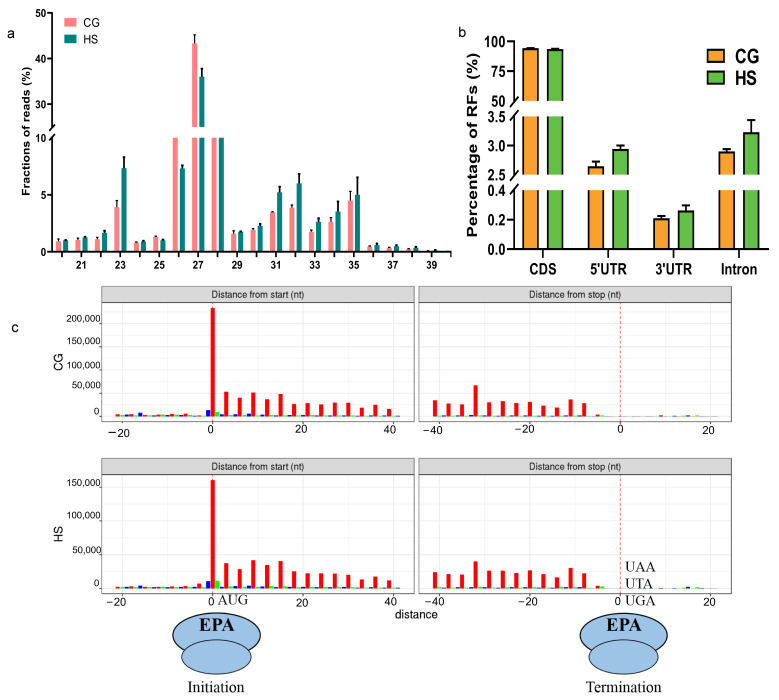
Analysis of Ribo-seq data in the CG and HS. (**a**) Comparison of RF length distribution between the CG and HS. (**b**) Distribution of RFs in coding sequences, the 5′UTR, and the 3′UTR in the CG and HS. (**c**) Total number of RFs along start and stop codon regions in the CG and HS datasets. The red, green, and blue bars show the reads aligned to the codons’ 1st, 2nd, and 3rd positions.

**Figure 2 ijms-25-08848-f002:**
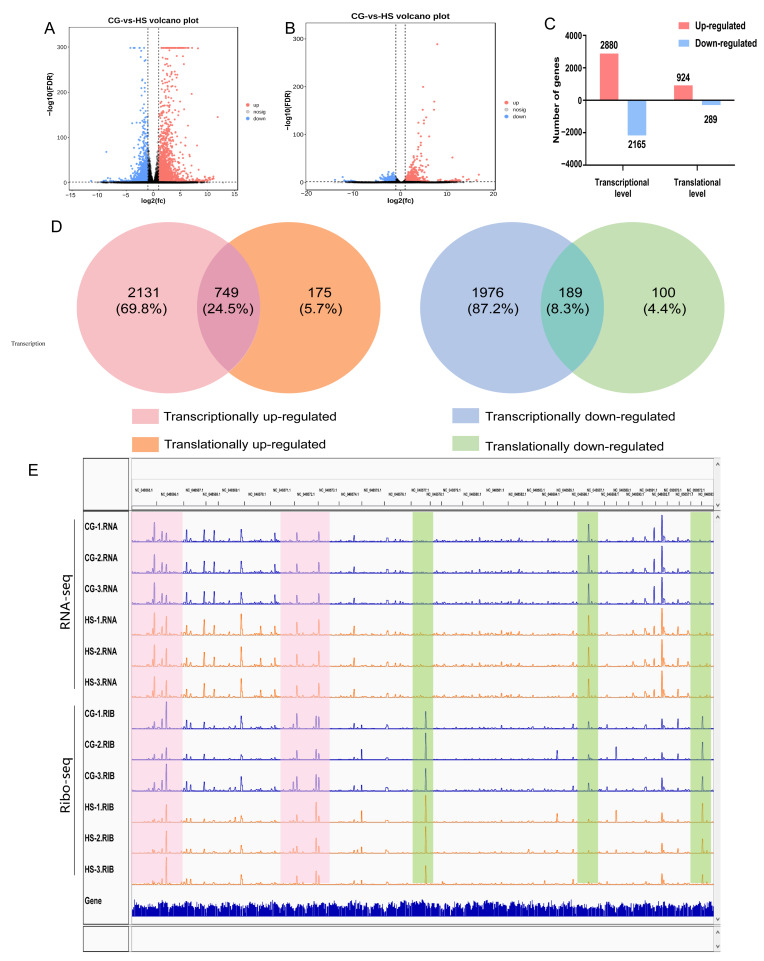
Heat-stress-induced transcriptional and translational responses. (**A**,**B**) Representation of differentially expressed genes (DEGs) in volcano plot format (|log2FC| > 1 and FDR < 0.05) during heat stress at transcriptional and translational levels. (**C**) Quantification of DEGs during heat stress at two levels. (**D**) Correlation between genes responsive to heat stress at two levels. Genes illustrated in (**D**) were obtained from (**C**). (**E**) Global RNA-seq and Ribo-seq tracks are shown in the IGV browser during heat stress. Alterations in gene expression levels due to heat stress are indicated by light-green shading. Noteworthy changes exclusively at the transcription or translation levels are displayed in light red.

**Figure 3 ijms-25-08848-f003:**
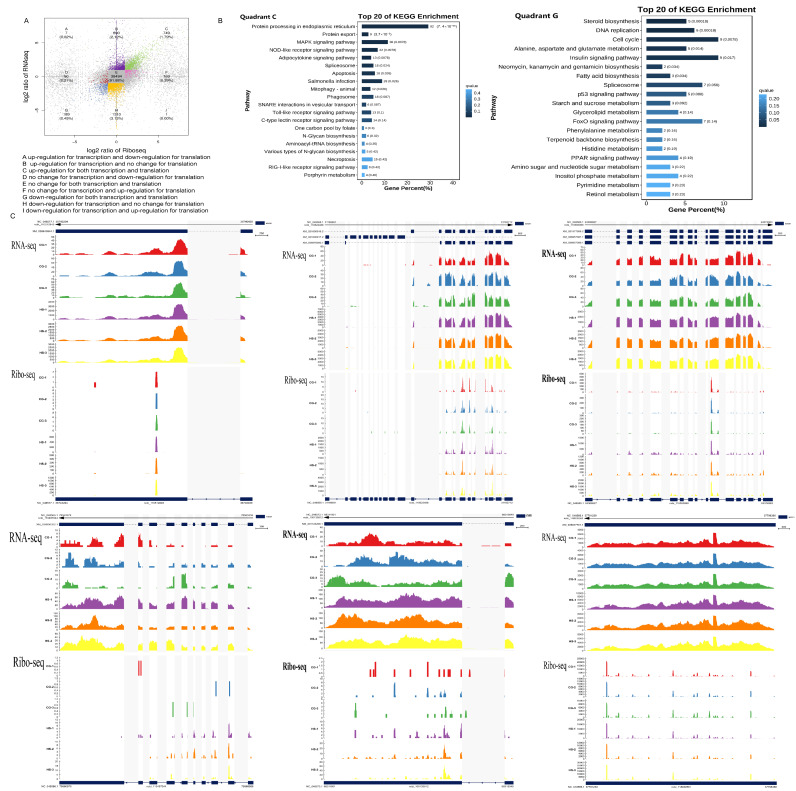
Transcriptional and translational modifications in rainbow trout liver under thermal stress were examined. (**A**) A scatter plot illustrates changes in gene expression at both transcriptional and translational levels after being exposed to heat stress. (**B**) Analysis of KEGG enrichment reveals the presence of heat-stress-responsive genes in quadrants C and G. (**C**) Visualization through the IGV browser illustrates the alterations in gene expression at both transcriptional and translational levels in rainbow trout following thermal stress, including genes like Hsp70 (ncbi_110512845), Hsp90a1 (ncbi_110522488), Hsp90b1 (ncbi_110500099), TLR5 (ncbi_100135812), C3-like (ncbi_118939581), and IRF6 (ncbi_110497044).

**Figure 4 ijms-25-08848-f004:**
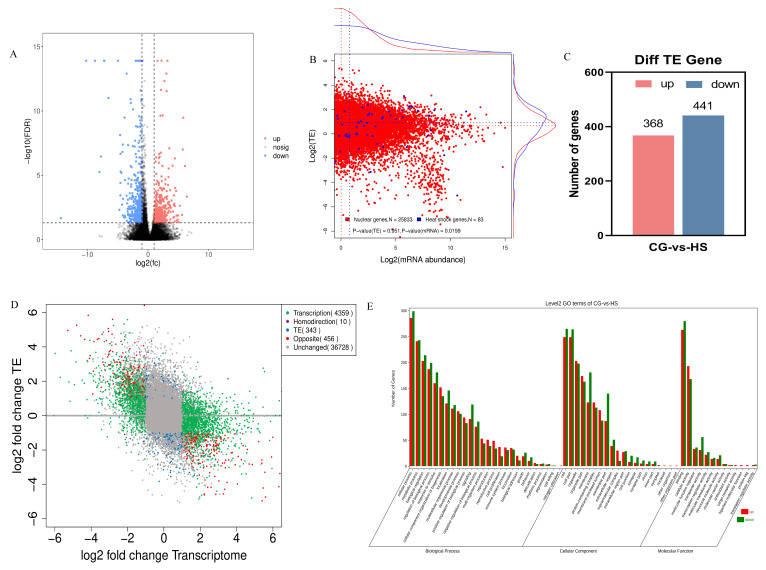
Analysis of TE across the genome in samples from the CG and HS. (**A**) A volcano plot representation of DEGs at the transcriptional and TE levels in response to heat stress. (**B**) Association between TE levels (log2) and mRNA abundance in HS samples, with concentration on the axes. Nuclear genes and heat stress genes are classified as red and blue dots, respectively, with average values indicated by dotted lines of the corresponding colors. The significance of differences was assessed through a one-tailed Student’s *t*-test. (**C**) Number of differentially expressed TE genes after heat stress. (**D**) A scatter plot illustrating changes in transcription and TE levels under heat stress, categorizing genes into five groups. (**E**) The results of gene ontology (GO) enrichment examination for differentially expressed TE after heat stress are provided.

**Figure 5 ijms-25-08848-f005:**
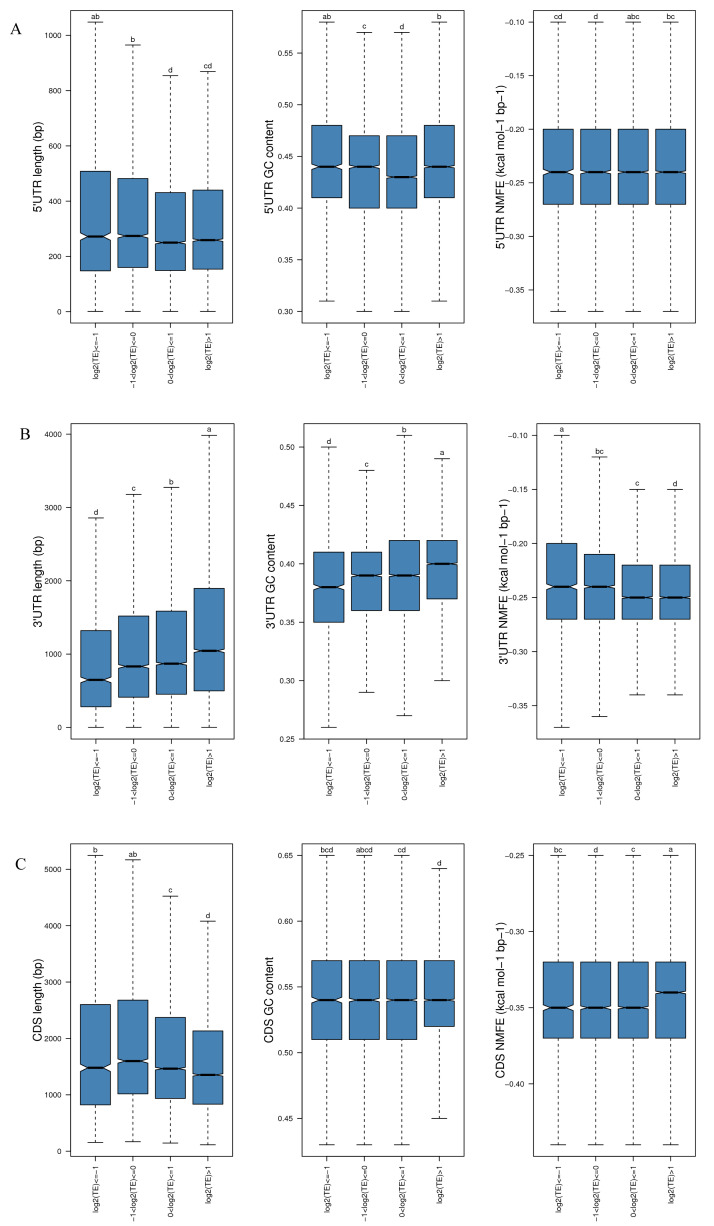
The influence of sequence traits on four types of TE in the (**A**) 3′UTR and (**B**) 5′UTR. (**C**) Coding sequences (CDSs) of samples from the HS. Labeled a–d show significant differences, determined by the x Student’s *t*-test with a significance level of *p* < 0.05.

**Figure 6 ijms-25-08848-f006:**
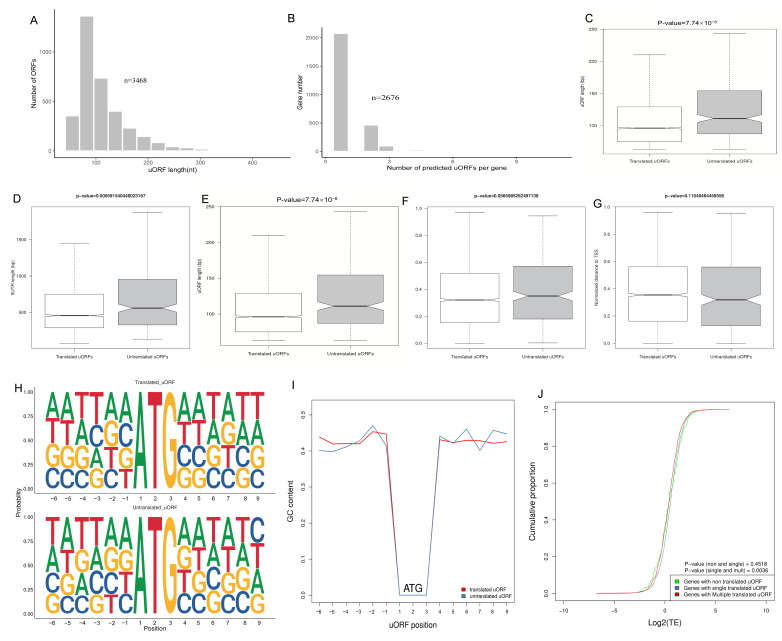
Characterization of uORFs in HS samples. (**A**) Distribution of predicted uORF lengths. (**B**) Predicted number of uORFs. (**C**) Comparison of the lengths of uORF sequences between translated and untranslated. (**D**) Comparison of the lengths of the 5′UTR between translated and untranslated. (**E**) Comparison of uORF NMFE between translated and untranslated. (**F**) Normalized distance to CDS start between translated and untranslated. (**G**) TSS between translated and untranslated. Student’s *t*-test was used to test p-values. (**H**) Kozak sequences of uORFs and main ORFs. (**I)** Comparing the GC content near the ATG initiation codon in translated and untranslated uORFs. (**J**) Examination of translation efficiency among genes featuring no, singular, and multiple translated uORFs. The *p*-value was determined using the Kolmogorov–Smirnov test.

## Data Availability

The datasets presented in this study can be found in online repositories (GSE263547, GSE262612).

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
