# Peer review of "Ribosome Profiling and RNA Sequencing Reveal Translation and Transcription Regulation under Acute Heat Stress in Rainbow Trout (Oncorhynchus mykiss, Walbaum, 1792) Liver"

_ijms, 2024, doi:10.3390/ijms25168848_

Round 1
Reviewer 1 Report
Comments and Suggestions for Authors
Zhao et al. explored the heat stress response in rainbow trout at both the transcriptional and translational levels. They found that post-transcriptional regulation is involved in the heat response. In particular, the expression pattern of some genes was similar at both the transcriptional and translational levels, implying their coordinated function across these levels. This study provides insights into the molecular mechanisms underlying heat stress response and highlights the importance of considering both transcriptional and translational regulation in understanding how organisms cope with environmental stress. However, some issues should be addressed.
1. Line 16-18, you observed that 32.8% of genes were expressed at both the transcriptional and translational levels, and you concluded that the transcriptional and translational responses to heat stress were independent. However, I believe that the co-expression of these genes at both levels suggests that they were interdependent, not completely independent. This interdependence may indicate that some responses to heat stress were coordinated across both transcriptional and translational levels.
2. Line 101, please point out the temperature used for the control group.
3. Line 102-103, how many individuals were sequenced?
4. Line 104-105, that is not a complete sentence.
5. Line 106, not only were Ribo-Seq raw reads preprocessed to remove low-quality sequences, but RNA-Seq raw reads were also processed in the same manner.
6. Line 107, R2, 2 should be in superscript.
7. Line 107-110, while the Pearson correlation coefficients (R²) between transcriptional and translational levels did increase from 0.82 to 0.83 under heat stress, the increase is quite small (only 0.01). This slight change may not be sufficient to conclusively indicate a significant increase in synchronization between transcription and translation. It would be beneficial to conduct further analysis or provide additional evidence to support the claim of increased synchronization under thermal stress conditions.
8. Figure 3C is confusing and makes it difficult to distinguish each gene.
9. Line 183-187, The first letter of the first word should be capitalized.
10. Line 158-160, “2.24% of the responding genes belonged to the corresponding groups (classes C and G) and that their expression was increased or decreased to a similar extent at the transcriptional and translational levels.” Line 211, “Only ten genes were regulated at the transcriptional and TEs levels.” Are there any hub genes between these two groups of genes? Perhaps you can conduct an in-depth analysis of the genes in these two groups, which might be helpful.
11. Line 212-215, What types of DEGs were used for the GO and KEGG analyses?
12. Line 226-238, When stating that something was longer or increased, you should specify what it is being compared to. Additionally, in the legend of Figure 5, what does CK represent? It would also be beneficial to show the gene characteristics of the CG and HS groups in the manuscript. Importantly, “the genes with lower TE (log2(TE) ≤ −1) had decreased NMFE in the 5'UTR region.” However, based on Figure 5, no significant differences were observed between the genes with lower TE (log2(TE) ≤ −1) and the other genes. A similar issue was noted for the GC content of the CDS.
13. Line 247-249, Figure 6A corresponds to 3,468 uORFs, while Figure 6B corresponds to 2,676 genes.
14. Line 258, what does CK group represent? It would be better to show Figure 6 C-G and Figure S4 A-E together in the manuscript.
15. Line 269-271, based on Figure 6H and 6I, it seems that the GC content in the sequences surrounding the start codons of translated uORFs and untranslated uORFs shows no remarkable difference.
16. Line 269-278, what about the characteristics of translated and untranslated uORFs in the control group? Are there differences between CG and HS groups? Please revise the discussion accordingly to include this information.
17. Line 314-323, the discussion contains too many descriptions of repeated results.
18. Line 394, 300 full-sci healthy what were obtained? What is the meaning of “full-sci”?
19. Line 402, what is the meaning of “liver samples were collected from [13]”? Why did you use liver samples for the subsequent analysis?
20. Line 403-404, “three biological replicates selected from CG and HS groups were subjected to Ribo-seq and RNA-seq”, finally, how many individuals were sampled from each group?
21. Line 412, line 435, please add the reference of the genome of rainbow trout.
22. Line 423-425, please pay attention to the units used for the reagents.
23. Line 456-458, what is the basis for dividing into these four categories? What does each category represent?
24. In the Results section, please move the discussion sentences to the Discussion section. It is not necessary to discuss them in the result section.
Comments on the Quality of English LanguageThe English is acceptable.
Author Response
Comments 1: Line 16-18, you observed that 32.8% of genes were expressed at both the transcriptional and translational levels, and you concluded that the transcriptional and translational responses to heat stress were independent. However, I believe that the co-expression of these genes at both levels suggests that they were interdependent, not completely independent. This interdependence may indicate that some responses to heat stress were coordinated across both transcriptional and translational levels.
Response 1: Thank you for pointing this out. We agree with this comment. Therefore, we have modified the sentence lines 18-19.
Comments 2: Line 101, please point out the temperature used for the control group.
Response 2: Thank you for pointing this out. We agree with this comment. Therefore, we have pointed out the temperature used for the control group line 114.
Comments 3: Line 102-103, how many individuals were sequenced?
Response 3: Thank you for pointing this out. We agree with this comment. Therefore, we have labelled three replicates per group for control and heat stress groups line 116.
Comments 4: Line 104-105, that is not a complete sentence.
Response 4: Thank you for pointing this out. We agree with this comment. Therefore, we have made the sentence complete lines 118-119.
Comments 5: Line 106, not only were Ribo-Seq raw reads preprocessed to remove low-quality sequences, but RNA-Seq raw reads were also processed in the same manner.
Response 5: Thank you for pointing this out. We agree with this comment. Therefore, we have added the RNA-seq line 120.
Comments 6: Line 107, R2, 2 should be in superscript.
Response 6: Thank you for pointing this out. We agree with this comment. Therefore, we have changed the R2 to R2 line 121.
Comments 7: Line 107-110, while the Pearson correlation coefficients (R²) between transcriptional and translational levels did increase from 0.82 to 0.83 under heat stress, the increase is quite small (only 0.01). This slight change may not be sufficient to conclusively indicate a significant increase in synchronization between transcription and translation. It would be beneficial to conduct further analysis or provide additional evidence to support the claim of increased synchronization under thermal stress conditions.
Response 7: Thank you for pointing this out. We agree with this comment. My description of this result is subjective and the expression has been adjusted, plus I want to emphasise that small changes in correlation coefficients may reflect long-term trends or cyclical changes. However, over time, these changes may accumulate and have significant biological effects lines 123-124.
Comments 8: Figure 3C is confusing and makes it difficult to distinguish each gene.
Response 8: Thank you for pointing this out. We agree with this comment. Therefore, we have improved clarity of Figure 3.
Comments 9: Line 183-187, The first letter of the first word should be capitalized.
Response 9: Thank you for pointing this out. We agree with this comment. Therefore, we have revised the first letter of the first word line 202.
Comments 10: Line 158-160, “2.24% of the responding genes belonged to the corresponding groups (classes C and G) and that their expression was increased or decreased to a similar extent at the transcriptional and translational levels.” Line 211, “Only ten genes were regulated at the transcriptional and TEs levels.” Are there any hub genes between these two groups of genes? Perhaps you can conduct an in-depth analysis of the genes in these two groups, which might be helpful.
Response 10: Thank you for pointing this out. We agree with this comment. Therefore, we have analysed there are some hub genes between these two groups of genes, such as hsp90a1 (ncbi_110522488) and hsp70a (ncbi_100135836). We have conducted in-depth analysis of the genes in these two groups in the discussion section.
Comments 11: Line 212-215, What types of DEGs were used for the GO and KEGG analyses?
Response 11: Thank you for pointing this out. We agree with this comment. Therefore, due to my unclear expression of the DEGs, which caused trouble for your review, I have revised the manuscript lines 227-228, 233.
Comments 12: Line 226-238, When stating that something was longer or increased, you should specify what it is being compared to. Additionally, in the legend of Figure 5, what does CK represent? It would also be beneficial to show the gene characteristics of the CG and HS groups in the manuscript. Importantly, “the genes with lower TE (log2(TE) ≤ −1) had decreased NMFE in the 5'UTR region.” However, based on Figure 5, no significant differences were observed between the genes with lower TE (log2(TE) ≤ −1) and the other genes. A similar issue was noted for the GC content of the CDS.
Response 12: Thank you for pointing this out. We agree with this comment. Therefore, we have revised and rewritten this section lines 248-260, 263-264.
Comments 13: Line 247-249, Figure 6A corresponds to 3,468 uORFs, while Figure 6B corresponds to 2,676 genes.
Response 13: Thank you for pointing this out. We agree with this comment. Therefore, we have changed the mistake lines 277-278.
Comments 14: Line 258, what does CK group represent? It would be better to show Figure 6 C-G and Figure S4 A-E together in the manuscript.
Response 14: Thank you for pointing this out. We agree with this comment. Therefore, we have changed the mistake, not CK, is CG line 288. Due to in the manuscript the Figure describes the heat stress group changes, the CG group figure is placed in the supplementary material, and we hope for your understanding!
Comments 15: Line 269-271, based on Figure 6H and 6I, it seems that the GC content in the sequences surrounding the start codons of translated uORFs and untranslated uORFs shows no remarkable difference.
Response 15: Thank you for pointing this out. We agree with this comment. Therefore, we have changed the mistake. I am very sorry for the trouble I caused you in reviewing the manuscript. Previously, due to a mistake in my work, I confused CG with HS, now have corrected the Figure 6H and 6I.
Comments 16: Line 269-278, what about the characteristics of translated and untranslated uORFs in the control group? Are there differences between CG and HS groups? Please revise the discussion accordingly to include this information.
Response 17: Thank you for pointing this out. We agree with this comment. Therefore, we have revised this section. In the control group, no significant change in the GC content of sequences around the start codon was observed in translated uORFs compared to untranslated uORFs, and there are no differences between CG and HS groups.
Comments 17: Line 314-323, the discussion contains too many descriptions of repeated results.
Response 17: Thank you for pointing this out. We agree with this comment. Therefore, we have revised this section lines 346-359.
Comments 18: Line 394, 300 full-sci healthy what were obtained? What is the meaning of “full-sci”?
Response 18: Thank you for pointing this out. We agree with this comment. Therefore, we have changed the mistake, 300 full-sib healthy rainbow trout were obtained. Full-sci is wrong, we have changed full-sci to full-sib line 455.
Comments 19: Line 402, what is the meaning of “liver samples were collected from [13]”? Why did you use liver samples for the subsequent analysis?
Response 19: Thank you for pointing this out. We agree with this comment. Therefore, we have changed the mistake, liver samples were collected from rainbow trout line 463. Our previous study found that the liver is the most important tissue for responding to thermal stimuli in rainbow trout. During heat stress, HSPs showed temporal and tissue specificity in different tissues, and compared with other tissues, HSPs in liver showed stronger heat sensitivity and the most significant up-regulation; during continuous chronic heat stress, the pathological changes of hepatocytes were gradually aggravated, and the damage was most obvious; in the metabolic pathway analysis of differentially expressed mRNAs before and after heat stress, many of biological processes involved in heat stress were related to the function of the liver. In the metabolic pathway analysis of differentially expressed mRNAs before and after heat stress, many biological processes involved in heat stress are related to the function of the liver, therefore, liver tissue is a more ideal detector of heat stress and a better material for the study of anti-heat stress mechanisms. The references to the following:
Li, Y. J., Huang, J. Q., Liu, Z., Zhou, Y. J., Xia, B. P., Wang, Y. J., Kang, Y. J., and Wang, J. F. (2017) Transcriptome analysis pro-vides insights into hepatic responses to moderate heat stress in the rainbow trout (Oncorhynchus mykiss). Gene. 619, 1-9.
Wang Y.N., Liu Z, Li Z, Shi H.N., Kang Y.J., Wang J.F., Huang J.Q., Jiang L. (2016) Effects of heat stress on respiratory burst, oxidative damage and SERPINH1 (HSP47) mRNA expression in rainbow trout (Oncorhynchus mykiss). Fish Physiol Biochem. 42, 701-10.
Comments 20: Line 403-404, “three biological replicates selected from CG and HS groups were subjected to Ribo-seq and RNA-seq”, finally, how many individuals were sampled from each group?
Response 20: Thank you for pointing this out. We agree with this comment. Therefore, we have redescribed this sentence lines 465-466, finally, three individuals were sampled from each group.
Comments 21: Line 412, line 435, please add the reference of the genome of rainbow trout.
Response 20: Thank you for pointing this out. We agree with this comment. Therefore, we have added the reference of the genome of rainbow trout line 474, line 498.
Comments 22: Line 423-425, please pay attention to the units used for the reagents.
Response 22: Thank you for pointing this out. We agree with this comment. Therefore, we have fixed the mistake on unit symbols lines 486-489.
Comments 23: Line 456-458, what is the basis for dividing into these four categories? What does each category represent?
Response 22: Thank you for pointing this out. We agree with this comment. Translational efficiency (TE) represents the proportion of total RNA molecules (usually referred to as mRNA) of a gene in a sample that binds to ribosomes and undergoes translation, and it is an important indicator for describing the RNA translation process. The mean values of the expression of the same gene at the translation and transcription levels in each group of samples were calculated separately, and then the TE of each gene in each group was calculated, TE = FPKMRibo-seq/FPKMRNA-seq. log2(TE) > 0 indicates that TE > 1, the post-transcription product of the gene accumulates and the RNA is utilised, log2(TE) < 0 is the opposite effect, log2 (TE) > 1 indicates that TE > 2, which means that the translation is more efficient, log2 (TE) < -1 is the opposite.
Comments 24: In the Results section, please move the discussion sentences to the Discussion section. It is not necessary to discuss them in the result section.
Response 24: Thank you for pointing this out. We agree with this comment. Therefore, we have moved the discussion sentences to the discussion section.

Reviewer 2 Report
Comments and Suggestions for Authors
This study, which comes from a skilled group on this topic, resulted in an interesting and fluent reading, focused on an innovative approach to a commercially important model species.
I suggest adding the eponym, Walbaum, 1792, to the title, abstract and first time mentioning the species in the main text.
I suggest avoiding the use of words already reported in the title among the keywords, please try to substitute them with other related ones.
The period between lines 41-52 is too auto-referenced, I understand your experience in this field, but enrich it with some other authors' studies. The sentence among lines 90-93 could result in unnecessary in this section.
All the figures in the result section are almost unreadable in the present form, so the reader is unable to evaluate them. I was able to revise them just from the supplementary material. Please solve this problem within the editorial office before the eventual publication.
Lines 423-425: please fix the mistake on unit symbols.
Try to slightly reduce the identity detected by the system.
Best regards
Author Response
Comments 1:I suggest adding the eponym, Walbaum, 1792, to the title, abstract and first time mentioning the species in the main text.
Response 1: Thank you for pointing this out. We agree with this comment. Therefore, we have added the Walbaum, 1792, to the title, abstract and first time mentioning the species in the main text lines 4, 10 and 44.
Comments 2:I suggest avoiding the use of words already reported in the title among the keywords, please try to substitute them with other related ones.
Response 2: Thank you for pointing this out. We agree with this comment. Therefore, we have replaced keywords with new relevant words line 30.
Comments 3:The period between lines 41-52 is too auto-referenced, I understand your experience in this field, but enrich it with some other authors' studies. The sentence among lines 90-93 could result in unnecessary in this section.
Response 3: Thank you for pointing this out. We agree with this comment. Therefore, we have cited some other authors' studies. And deleted 90-93 unnecessary in this section.
Comments 4:All the figures in the result section are almost unreadable in the present form, so the reader is unable to evaluate them. I was able to revise them just from the supplementary material. Please solve this problem within the editorial office before the eventual publication.
Response 4: Thank you for pointing this out. We agree with this comment. Therefore, we have changed sharpness of all the figures in the result section.
Comments 5:Lines 423-425: please fix the mistake on unit symbols.
Response 5: Thank you for pointing this out. We agree with this comment. Therefore, we have fixed the mistake on unit symbols lines 486-489.

Reviewer 3 Report
Comments and Suggestions for Authors
Author Response
General comments:
- The title should be revised to reflect the study's aim more clearly.
Response 1: Thank you for pointing this out. We agree with this comment. Therefore, we have changed the title lines 2-5.
- The authors should avoid using pronouns such as “we”, “our” and “us” in the whole text.
Response 2: Thank you for pointing this out. We agree with this comment. Therefore, we have avoided using pronouns such as “we”, “our” and “us” in the whole text.
- The paper needs revision to improve the language, sentence construction, and easy comprehension.
Response 3: Thank you for pointing this out. We agree with this comment. Therefore, we have changed the language, sentence construction to make it easier to understand in the whole text.
- Please ensure that all heading and sub-heading numbering is verified and revised as needed.
Response 4: Thank you for pointing this out. We agree with this comment. Therefore, we have changed all heading and sub-heading numbering.
Abstract:
- In the Abstracts, the main aim of the study should be mentioned more clearly.
Response 5: Thank you for pointing this out. We agree with this comment. Therefore, we have revised the aim of the study more clearly lines 24-27.
- Please specify the originality and value of the research and its contribution to international
literature.
Response 6: Thank you for pointing this out. We agree with this comment. Therefore, we have illustrated the originality and value of the research and its contribution to the international literature lines 24-27.
Keywords:
- Avoid using keywords already mentioned in the title (e.g., Rainbow trout) and replace them
with new relevant words in the text.
Response 7: Thank you for pointing this out. We agree with this comment. Therefore, we have replaced keywords with new relevant words line 30.
Introduction:
- Page 1, line 41: Some of these studies should be added after “In previous studies”.
Response 8: Thank you for pointing this out. We agree with this comment. Therefore, we have added some of these studies after “In previous studies” line 47.
- Page 2, lines 85-86: The statement “In this study, we comprehensively and systematically investigated the gene expression changes in rainbow trout under acute heat stress using RNA-seq and Ribo-seq.” should be revised and rewritten.
Response 9: Thank you for pointing this out. We agree with this comment. Therefore, we have revised and rewritten this section lines 92-96, 104-106.
10.The novelty of this study can be further justified by highlighting the main contributions to the existing literature. This should be presented in the literature review related to the work.
Response 10: Thank you for pointing this out. We agree with this comment. Therefore, we have revised and rewritten this section lines 92-96, 104-106.
11.Consider adding the significance of the study in the Introduction section.
Response 11: Thank you for pointing this out. We agree with this comment. Therefore, we have added the significance of the study in the last paragraph of the introduction lines 104-106.
12.The authors should highlight how the proposed study addresses the existing gaps.
Response 12: Thank you for pointing this out. We agree with this comment. Therefore, we have highlighted how the proposed study addresses the existing gaps. In previous studies, most studies have focused on the biochemical level, gene expression, proteomics, and metabolomics, but lacked studies at the translational level. Therefore, the present study fills the gap in existing studies on heat stress in rainbow trout by providing a comprehensive view of molecular responses at the transcriptional and translational levels lines 56-58, 92-96.
13.Please explain why it is timeliness to explore such a study and what makes it different from other studies.
Response 13: Thank you for pointing this out. We agree with this comment. Therefore, we have explained why it is timeliness to explore such a study and what makes it different from other studies. The exploration of translational regulation in rainbow trout under thermal stress is timely due to the increasing global temperatures affecting aquatic ecosystems. As climate change intensifies, understanding how fish adapt at the molecular level is crucial for developing strategies to mitigate the impact on fish populations and aquaculture. And by focusing on the translational level, our research offers a novel perspective that complements the extensive body of work on transcriptional regulation. This approach is particularly relevant as it has the potential to uncover post-transcriptional regulatory mechanisms that may be key to the rapid adaptation of fish to thermal stress.
14.The objectives/research questions should be added to the end of the Introduction section.
Response 14: Thank you for pointing this out. We agree with this comment. Therefore, we have added the objectives of the study in the last paragraph of the introduction.
Results:
15.Ensure all figures and tables are cited and discussed in the main text.
Response 15: Thank you for pointing this out. We agree with this comment. Therefore, we have ensured all figures and tables are cited and discussed in the main text.
16.Page 4, line 142: The full term "IGV" should be added on first use.
Response 14: Thank you for pointing this out. We agree with this comment. Therefore, we have added the the full term "IGV" lines 159-160.
Discussion:
17.The Discussion section should highlight the following requirements:
- a) State the main results of the study (do not repeat the inputs of the results section).
Response a: Thank you for pointing this out. We agree with this comment. Therefore, we have revised the discussion.
- b) Compare and interpret these results in detail with the findings of recent studies.
Response b: Thank you for pointing this out. We agree with this comment. Therefore, we have compared and interpreted these results in detail with the findings of recent studies.
- c) At the end of this section, add the main limitations of the study.
Response c: Thank you for pointing this out. We agree with this comment. Therefore, we have added the main limitations of the study at the end of discussion.
Materials and method:
18.More details are needed regarding the methods that were used.
Response 18: Thank you for pointing this out. We agree with this comment. Therefore, we have added a more detailed description.
Conclusion:
19.Consider revising the Conclusion as follows:
- a) The focus should be on restating the main results and demonstrating how the research questions have been thoroughly examined and explained.
Response a: Thank you for pointing this out. We agree with this comment. Therefore, we have revised and rewritten the conclusion.
- b) It is important to enrich this section with paragraphs discussing the international policy implications of the study's findings.
Response b: Thank you for pointing this out. We agree with this comment. Therefore, we have revised and rewritten the conclusion.
- c) The theoretical and practical implications of the study's findings should be highlighted in this section.
Response c: Thank you for pointing this out. We agree with this comment. Therefore, we have revised and rewritten the conclusion.
References:
20.There are a lot of old references (before 2015) in the whole text. The Author/s should be tried to update them (2020-2024).
Response 20: Thank you for pointing this out. We agree with this comment. Therefore, we have updated the old references.

Reviewer 4 Report
Comments and Suggestions for Authors
The manuscript provides an interesting addition to the knowledge of the heat stress response in rainbow trouts. I have the following minor comments for improvement:
-Title, Abstract, Manuscript body: Species and genus authorities should be added upon first mentioning of a species / genus name throughout the manuscript. Depending on the journal, this might be done without indicating the year of the original description (usually the reference is not cited in the list of references then) or with indication of the year of the original description, with the reference to be included in or excluded from the list of references.
-Lines 32–33: Higher than what? And a reference is needed here as certain warm water fishes are also sensitive to temperature increases.
-Last paragraph of Introduction: This paragraph should provide the study objectives. However, it is written more like the beginning of a Discussion or a Conclusion section. Accordingly, this paragraph needs to be rewritten to include only the study objectives.
-It is unusual to place the Method and Methods section behind the Results and Discussion sections. Is this order in line with the journal’s instructions for authors?
-The figures are hardly readable. The quality must be improved and it might be necessary to enlarge details such as font size. And please also check for spelling errors such as a double space character in Figure 3, A: “B up-regulation…”
Author Response
Comment 1: Title, Abstract, Manuscript body: Species and genus authorities should be added upon first mentioning of a species / genus name throughout the manuscript. Depending on the journal, this might be done without indicating the year of the original description (usually the reference is not cited in the list of references then) or with indication of the year of the original description, with the reference to be included in or excluded from the list of references.
Response 1: Thank you for pointing this out. We agree with this comment. Therefore, we have added the Walbaum, 1792, to the title, abstract and first time mentioning the species in the main text lines 4, 10 and 44.
Comment 2: Lines 32–33: Higher than what? And a reference is needed here as certain warm water fishes are also sensitive to temperature increases.
Response 2: Thank you for pointing this out. We agree with this comment. Therefore, we have changed the sentence line 39.
Comment 3: Last paragraph of Introduction: This paragraph should provide the study objectives. However, it is written more like the beginning of a Discussion or a Conclusion section. Accordingly, this paragraph needs to be rewritten to include only the study objectives.
Response 3: Thank you for pointing this out. We agree with this comment. Therefore, we have rewritten this paragraph lines 92-96, 104-106.
Comment 4: It is unusual to place the Method and Methods section behind the Results and Discussion sections. Is this order in line with the journal’s instructions for authors?
Response 4: Thank you for pointing this out. This order matches the journal's requirements for authors.
Comment 5: The figures are hardly readable. The quality must be improved and it might be necessary to enlarge details such as font size. And please also check for spelling errors such as a double space character in Figure 3, A: “B up-regulation…”
Response 5: Thank you for pointing this out. We agree with this comment. Therefore, we have improved the clarity of all figures.

Round 2
Reviewer 1 Report
Comments and Suggestions for Authors
1. You mentioned that Ribo-seq is a novel technology/approach, while I think it is not novel.
2. Line 123-124, this sentence should be moved to the discussion.
3. Line 152, the effects ….. were investigated.
4. Regarding comments 23, you can put the explanation to the 4.7 section.
5. Line 219-222, “Based on the changes in gene TE and transcription levels, the genes were categorized into five groups. Only then genes were regulated at the transcriptional and TEs levels, such as XXXX.” The genes mean what kind of genes?
Comments on the Quality of English Languageacceptable
Author Response
Comments 1: You mentioned that Ribo-seq is a novel technology/approach, while I think it is not novel.
Response 1: Thank you for pointing this out. We agree with this comment. Therefore, after further literature research and team discussions, we realised that the Ribo-seq technique had become a routine method in certain research areas, therefore, we corrected the mistake lines 78-81.
Comments 2: Line 123-124, this sentence should be moved to the discussion.
Response 2: Thank you for pointing this out. We agree with this comment. Therefore, we have moved this sentence to the discussion lines 369-371.
Comments 3: Line 152, the effects ….. were investigated.
Response 3: Thank you for pointing this out. We agree with this comment. Therefore, we have modified this sentence lines 162-163.
Comments 4: Regarding comments 23, you can put the explanation to the 4.7 section.
Response 4: Thank you for pointing this out. We agree with this comment. Therefore, we have put the explanation to the 4.7 section lines 547-557.
Comments 5: Line 219-222, “Based on the changes in gene TE and transcription levels, the genes were categorized into five groups. Only then genes were regulated at the transcriptional and TEs levels, such as XXXX.” The genes mean what kind of genes?
Response 5: Thank you for pointing this out. We agree with this comment. Therefore, we have added the genes lines 236-237.

Reviewer 3 Report
Comments and Suggestions for Authors
Title of the manuscript: Ribosome profiling reveals a dynamic translational landscape in
rainbow trout (Oncorhynchus mykiss) liver under acute heat stress
Manuscript ID: ijms-3098414
The authors have adequately addressed my previous comments but there are still some issues that need to be resolved. Here are my minor comments:
1. Page 1, lines 13-14: Consider revising this statement "we this study performed ribosome profiling assay" to "In this study, we performed a ribosome profiling assay".
2. On page 1, line 19, the statement "demonstrating that heat stress were coordinated across both transcriptional and translational levels" can be revised to "demonstrating that heat stress is coordinated across both transcriptional and translational levels."
3. On page 3, lines 57-59: This statement "Existing studies have not yet delved into the effects of heat stress on translation regulation in rainbow trout" is not clear enough. Please cite some specific studies that have investigated related aspects, and then highlighting the gap this study addresses.
4. some findings are mentioned several times in different sections of the manuscript without adding new information that need more attention and revision.
5. The Conclusion section should strongly highlight the new contributions of the study and its potential impact on aquaculture practices. At the moment it feels more like a summary of findings than a strong conclusion that ties everything together.

Moderate editing of English language required
Author Response
Comments 1:Page 1, lines 13-14: Consider revising this statement "we this study performed ribosome profiling assay" to "In this study, we performed a ribosome profiling assay".
Response 1: Thank you for pointing this out. We agree with this comment. Therefore, we have modified the sentence lines 13-14.
Comments 2:On page 1, line 19, the statement "demonstrating that heat stress were coordinated across both transcriptional and translational levels" can be revised to "demonstrating that heat stress is coordinated across both transcriptional and translational levels."
Response 2: Thank you for pointing this out. We agree with this comment. Therefore, we have modified the sentence lines 19-20.
Comments 3:On page 3, lines 57-59: This statement "Existing studies have not yet delved into the effects of heat stress on translation regulation in rainbow trout" is not clear enough. Please cite some specific studies that have investigated related aspects, and then highlighting the gap this study addresses.
Response 3: Thank you for pointing this out. We agree with this comment. Therefore, we have cited some specific studies that have investigated related aspects, and then highlighting the gap this study addresses lines 58-67.
Comments 4:some findings are mentioned several times in different sections of the manuscript without adding new information that need more attention and revision.
Response 4: Thank you for pointing this out. We agree with this comment. Therefore, we have modified the mistake in the manuscript.
Comments 5:The Conclusion section should strongly highlight the new contributions of the study and its potential impact on aquaculture practices. At the moment it feels more like a summary of findings than a strong conclusion that ties everything together.
Response 5: Thank you for pointing this out. We agree with this comment. Therefore, we have modified the conclusion lines 571-581.
